# Assessing trade-offs in avian behaviour using remotely collected data from a webcam

**Kevin A. Wood**[1], **Rebecca Lacey**[2], **Paul E. Rose**[1,2]*

**1** Wildfowl & Wetlands Trust, Slimbridge Wetland Centre, Slimbridge, Gloucestershire, United Kingdom,
**2** Centre for Research in Animal Behaviour, Psychology, Washington Singer, University of Exeter, Exeter, Devon, United Kingdom

* p.rose@exeter.ac.uk

## Abstract

Individual animals engage in many behaviours which are mutually exclusive, and so where individuals increase the duration of time spent on one type of behavioural activity, this must be offset by a corresponding decrease in at least one other type of behaviour. To understand the variation observed in animal behaviour, researchers need to know how individuals trade-off these mutually-exclusive behaviours within their time-activity budget. In this study, we used remotely collected behavioural observations made from a live-streaming webcam to investigate trade-offs in the behaviour of two bird species, the mute swan (*Cygnus olor*) and whooper swan (*Cygnus cygnus*). For both species, we tested for correlations in the duration of time spent on key mutually exclusive behaviours: aggression, foraging, maintenance, and resting. We detected a negative association between aggression and resting behaviours in both species, indicating that increased aggression is achieved at the expense of resting behaviour. In contrast, there was no apparent trade-off between aggression and foraging, aggression and maintenance, or maintenance and resting. Foraging and resting behaviours were negatively correlated in both species, highlighting a trade-off between these distinct modes of behaviour. A trade-off between foraging and maintenance behaviours was detected for the sedentary mute swans, but not the migratory whooper swans. Our findings show how birds can trade-off their time investments in mutually exclusive behaviours within their time-activity budgets. Moreover, our study demonstrates how remotely-collected data can be used to investigate fundamental questions in behavioural research.

## Introduction

Many of the behaviours exhibited routinely by animals are mutually exclusive [1]. For example, an individual animal cannot forage and rest at the same time. Individuals therefore face key decisions about which behaviours to engage in at any given time. Such decision-making is expected to lead to trade-offs in the types of behaviour exhibited by individuals [1, 2]. Where individuals increase the duration of time spent on one type of behavioural activity, this must

**Data Availability Statement:** Both the data files and the analytical code used in our study can be accessed via the following DOI: https://doi.org/10.6084/m9.figshare.20063225.

**Funding:** The author(s) received no specific funding for this work.

**Competing interests:** The authors have declared that no competing interests exist.

be offset by a decrease in one or more other types of behaviour [2–4]. As such, the behaviours that an individual exhibits represent a zero-sum game.

Behavioural researchers routinely use time-activity budgets to describe the relative amounts of time that animals spend engaged in different behaviours [5–7]. Time-activity budgets can therefore be a powerful tool for assessing differences in the behaviour of different animal species, as well as differences in behaviour within animal species [8]. The literature on time-activity budgets recorded for birds show that the time investment made on different behaviours can be highly variable, even for concurrent observations of different birds within the same population in a shared habitat [8, 9]. At least some of this variation within a population may reflect trade-offs in behavioural activities, as individuals prioritise certain behaviours over others, depending on their state and environmental conditions [10]. To improve our understanding of the variation observed in animal behaviour, researchers require knowledge of how animals may trade-off mutually-exclusive behaviours within their time-activity budget. In particular, information is needed on the degree to which the different behaviours displayed by a species are traded off against each other.

In this study, we used remote behavioural observations made via a live-streaming webcam to investigate the trade-offs in avian behaviour. Webcams have become a valuable tool in the study of animal behaviour, as they allow behavioural observations to be made without disturbance to focal individuals and without the need for researchers to be physically present at study sites [11–15]. We selected two common waterbirds as our focal species, the mute swan (*Cygnus olor*) and whooper swan (*Cygnus cygnus*). Their large body size, distinctive white plumage, and use of open-water habitats have allowed researchers to make detailed behavioural observations [9, 13, 16–20].

Where the two species co-exist, mute and whooper swans show similar, but not identical, patterns of behaviour [9]. Among swans, foraging, resting, and maintenance behaviours such as preening are the most commonly observed activities [21], although the amount of time spent on these behaviours shows marked variation between studies [9, 18–20]. Other key behaviours include aggression towards conspecifics and heterospecifics, although previous studies have found high levels of variation in how much time is spent on aggression between individual swans [9, 13] and between swan populations [31]. Swans cannot be engaged in aggressive interactions such as threat displays, pecking, or striking with their wings, at the same time as behaviours such as foraging, maintenance or resting [16–20, 22]. The prominence of foraging, resting, and maintenance behaviours within the time-activity budgets of swans [21] suggests that as additional behaviours such as aggression are undertaken, these more common behaviours might be involved in any trade-offs. However, the extent to which swans trade-off different behavioural activities within their time-activity budgets, has received little attention from researchers.

As aggression is a means of acquiring and maintaining access to food resources [23–25], we expected that individuals who were motivated to forage would engage in aggressive interactions with other individuals. However, because aggression and foraging are mutually exclusive types of behaviour, over short time durations we would not expect a positive association between these two behaviours either. Our first prediction was therefore that we would not observe a correlation between the time spent on aggressive and foraging behaviours. Instead, we expected birds to gain the time spent on aggression from other behaviours that can be deferred, such as maintenance and resting behaviours. Our second and third predictions were therefore that higher durations spent on aggression would be associated with lower durations spent on resting and maintenance behaviours, respectively. Finally, our fourth prediction was that we would observe trade-offs (i.e. a negative statistical association) between all non-

aggressive behaviours (e.g. foraging, maintenance, and resting), as these activities represent distinct modes of behaviour [1].

## Methods

### Study system

The Wildfowl & Wetland Trust (WWT) Caerlaverock reserve (54˚59´2.4˝ N, 3˚30´0˝ W) in southwest Scotland is an important site for wintering waterbirds, including both mute and whooper swans [13, 16, 26]. The 587 ha site comprises a patchwork of aquatic and terrestrial habitats, including small lakes used by waterbirds for feeding and roosting. The non-migratory mute swans are resident at Caerlaverock throughout the year, whilst whooper swans are winter visitors present at Caerlaverock between October and March [16]. Both swan species are known to use the habitats within the reserve for all behaviours within their ethograms, including foraging, roosting and maintenance behaviours, as well as aggressive interactions with conspecifics and heterospecifics [9, 16]. Tracking studies of GPS-tagged individuals indicated that swans fed and roosted on the reserve, as many remained there for extended periods of time [27]. Nationally, the mute swan population has been stable since c.2000, whilst whooper swan numbers have increased steadily in recent decades [28, 29]. Within the Solway Estuary (which includes the WWT Caerlaverock reserve) the mean of the peak winter counts of individual swans between 2015/2016 and 2019/2020 was 76 mute swans (range 73–83) and 303 whooper (range 160–487) [26]. Previous research has found a balanced sex ratio among swans overwintering at our study site [16]. Whooper swan family groups arrive together and typically remain together during winter [30], although this behaviour is somewhat less common among mute swans, for which family groups may break up sooner [31].

### Data collection

To quantify swan behaviour, we used a focal sampling approach, *sensu* [32], in which individual swans were observed for 15 minute periods. An observation period of 15 minutes was therefore selected as a duration for which it was practical to follow a single individual, and made our approach comparable with those of earlier studies of swan behaviour which used similar durations of observations [9, 17, 20, 33, 34]. During each observation, the number of seconds spent by the focal swan on each of four mutually exclusive behaviours was recorded: aggression, foraging, maintenance, and resting. Aggressive behaviours include threat displays, as well as striking at opponents with wings or beak, and are commonly observed among birds [35–37]. Foraging referred to all behaviours related to seeking, acquiring, and consuming food, maintenance behaviours included all preening, stretching and comfort activities, whilst resting represented periods of apparent inactivity such as sleeping [9]. Any time for which the swan was not visible during the observation period, for example, if it was obscured by other individuals, was not included in the calculations of the time spent on behaviours. The time spent on each behaviour was expressed as a proportion of the total time for which the swan was visible during the observation period. The species identity of the focal individual was recorded during each observation; the two species could be distinguished readily during observations due to interspecific differences in the dominant colouration of the bill (orange and yellow for mute and whooper swans, respectively) and their body posture (mute swans typically hold their neck in a characteristic s-shaped curve, whereas whooper swans typically hold their neck straight) [30, 31].

In order to select focal individuals at random, and thus avoid any observer bias, we superimposed a numbered 18 x 10 grid over the webcam image; each grid cell was assigned an unique number, and we used a random number generator to select a grid cell. Of the swans

within that cell, the individual closest to the centre of the grid cell was selected as the focal individual for that 15 minute observation period. If the selected grid cell did not contain any swans, the process was repeated until an occupied cell was selected. This allowed us to randomise the selection of individuals, as well as the swan species that was selected. As there were markedly fewer juveniles at the study sites, our observations were made of adult birds. As the two swan populations were observed to use all grid cells, as well as all parts of the study lake, we see no reason why any individual would have been sampled more frequently than others.

A total of 119 observations were carried out between mid-November 2020 and mid-March 2021, inclusive (November $n = 9$, December $n = 22$, January $n = 41$, February $n = 32$, March $n = 15$). This sampling pattern was organised to obtain data across the months during which both species of swans were present at the site [13, 16, 34]. To obtain data from across the diurnal active period of the swans, observations were made only at 09:30, 12:00, 12:30, and 14:30. Overall, behavioural observations were made for a total of 119 swans (81 mute swans and 38 whooper swans). These observations comprising 1,448 minutes of observations (mute swans = 983 minutes, whooper swans = 465 minutes).

All behavioural observations were made remotely via a live-streaming webcam (AXIS Q6035-E PTZ Dome Network Camera), which was fixed in place facing directly outwards over the main roosting and feeding lake used by the swans. The angle and magnification were not adjusted during the study so that the field-of-view was standardised across all observation periods. Further information on the webcam set up is available in earlier published work [13].

## Ethics

This observational study was carried out with the prior approval of the ethics committee of the College of Life and Environmental Sciences of the University of Exeter (eCLESPsy002195). Our observational study of non-human animals did not feature human participants and so participant consent was not required. As data collection was conducted virtually via a publically-accessible live-streaming webcam, no physical visits to the study site were undertaken, and hence no study site permits were required.

## Statistical analyses

All statistical analyses were carried out in R version 3.6.3 [38]. Prior to assessing correlations between behaviours, we first tested whether either swan species showed marked differences in behaviour across the four times of day at which we made our observations. We therefore used Kruskal-Wallis tests, implemented using the *kruskal.test* function in R, to determine whether there were statistically significant differences in the time spent by each swan species on each behaviour (aggression, maintenance, foraging, resting) between each of the four time periods at which we made our behavioural observations. As we were interested in differences between any time period, we modelled the time periods as factors rather than as a continuous variable. We selected the non-parametric Kruskal-Wallis tests as the presence of zeros in our data set (e.g. where behaviours were not observed for some focal individuals) meant that the assumption of parametric tests such as Analysis of Variance (ANOVA) could not be met [39]. The *P* values were adjusted using Holm-Bonferroni corrections, implemented via the *p.adjust* function in R, to account for multiple comparisons [40].

We subsequently ran pairwise Kendall's $\tau$ correlations [41] using the *cor.test* function in R to test the strength and direction of the association between the proportions of time spent on each behaviour: aggression vs foraging, aggression vs maintenance, aggression vs resting, foraging vs maintenance, foraging vs resting, and maintenance vs resting. Again, we selected the non-parametric Kendall's $\tau$ correlations as the presence of zeros in our data set meant that the

assumption of parametric correlation analyses such as Pearson's *r* could not be met [39]. Kendall's *τ* values range from -1.0 (a perfect negative association), to +1.0 (a perfect positive association), with a value of 0.0 indicating no association between two variables [41]. As we knew *a priori* that the two swan species exhibit differences in behaviours [9, 13], we ran the correlation analyses separately for each species. Furthermore, as previous research on the swans at WWT Caerlaverock had shown that behaviour did not vary markedly with time of day [13], data from all observation periods were combined for the analyses; the validity of this assumption was tested using the Kruskal-Wallis tests previously described. Statistically significant positive or negative associations between the proportions of time spent on behaviours were attributed where $P < 0.05$, after $P$ values had been adjusted using Holm-Bonferroni corrections, implemented via the *p.adjust* function in R, to account for multiple comparisons [40].

## Results

Mute swans spent the highest proportion of their time engaged in foraging behaviour (mean ± 95% CI = 0.421 ± 0.043), and the lowest proportion of their observed time on aggressive interactions (0.141 ± 0.027; Fig 1). Mute swans spent intermediate proportions of their time-activity budget on resting (0.184 ± 0.036) maintenance behaviours (0.254 ± 0.030; Fig 1).

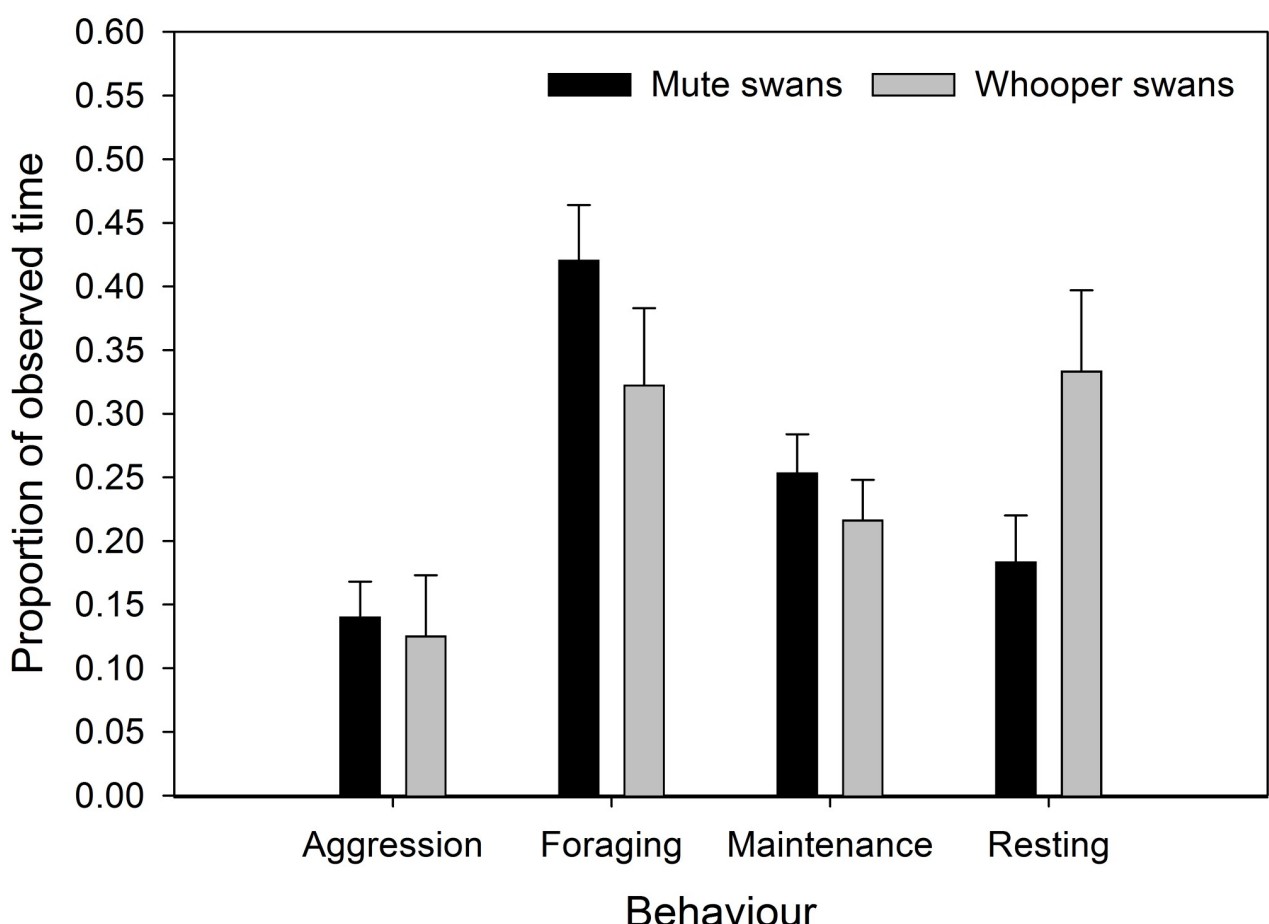

**Fig 1. Time-activity budgets.** The mean (±95% CI) proportion of observed time that mute swans (black bars; n = 81) and whooper swans (grey bars; n = 38) spent on each behaviour.

In contrast, whooper swans spent more time resting (0.334 ± 0.067) than on any other behaviour (Fig 1). Similar to mute swans, the whooper swans spent the lowest proportion of their observed time on aggressive interactions (0.126 ± 0.047; Fig 1). Foraging and resting accounted for similar proportions of their time-activity budget, 0.323 ± 0.060 and 0.334 ± 0.063, respectively (Fig 1).

Our analyses detected no statistically significant differences in the time spent on each behaviour by mute swans between the four time periods at which we made our observations: aggression ($\chi^2$ = 1.81, d.f. = 3, $P$ = 1.000; Fig 2a); maintenance ($\chi^2$ = 5.14, d.f. = 3, $P$ = 0.973;

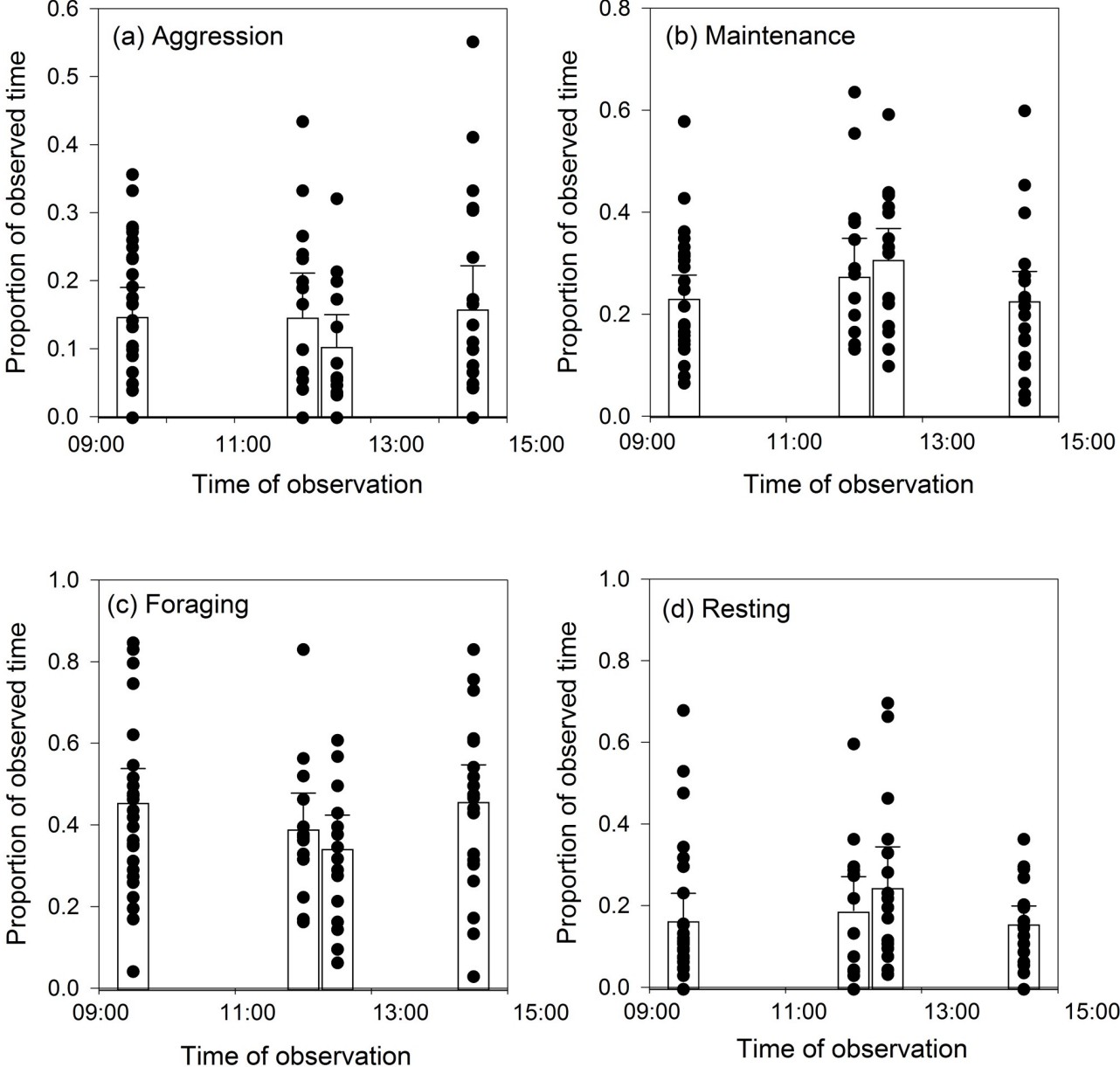

**Fig 2. Observed mute swan behaviour at different times of day.** The bars represent the mean ± 95% CI time spent on each behaviour, whilst the filled circles represent the individual data points.

Fig 2b); foraging ($\chi^2$ = 4.12, d.f. = 3, $P$ = 0.996; Fig 2c); resting ($\chi^2$ = 2.48, d.f. = 3, $P$ = 1.000; Fig 2d).

We found no statistically significant differences in the time spent on each behaviour by whooper swans between the four time periods at which we made our observations: aggression ($\chi^2$ = 2.33, d.f. = 3, $P$ = 1.000; Fig 3a); maintenance ($\chi^2$ = 7.63, d.f. = 3, $P$ = 0.435; Fig 3b); foraging ($\chi^2$ = 7.28, d.f. = 3, $P$ = 0.444; Fig 3c); resting ($\chi^2$ = 4.90, d.f. = 3, $P$ = 0.973; Fig 3d).

Among mute swans, the proportion of time spent on aggression was negatively correlated with the proportion of time spent resting ($\tau$ = -0.225, $P$ = 0.036; Fig 4). Similarly, we also detected negative correlations between the proportion of time engaged in foraging and

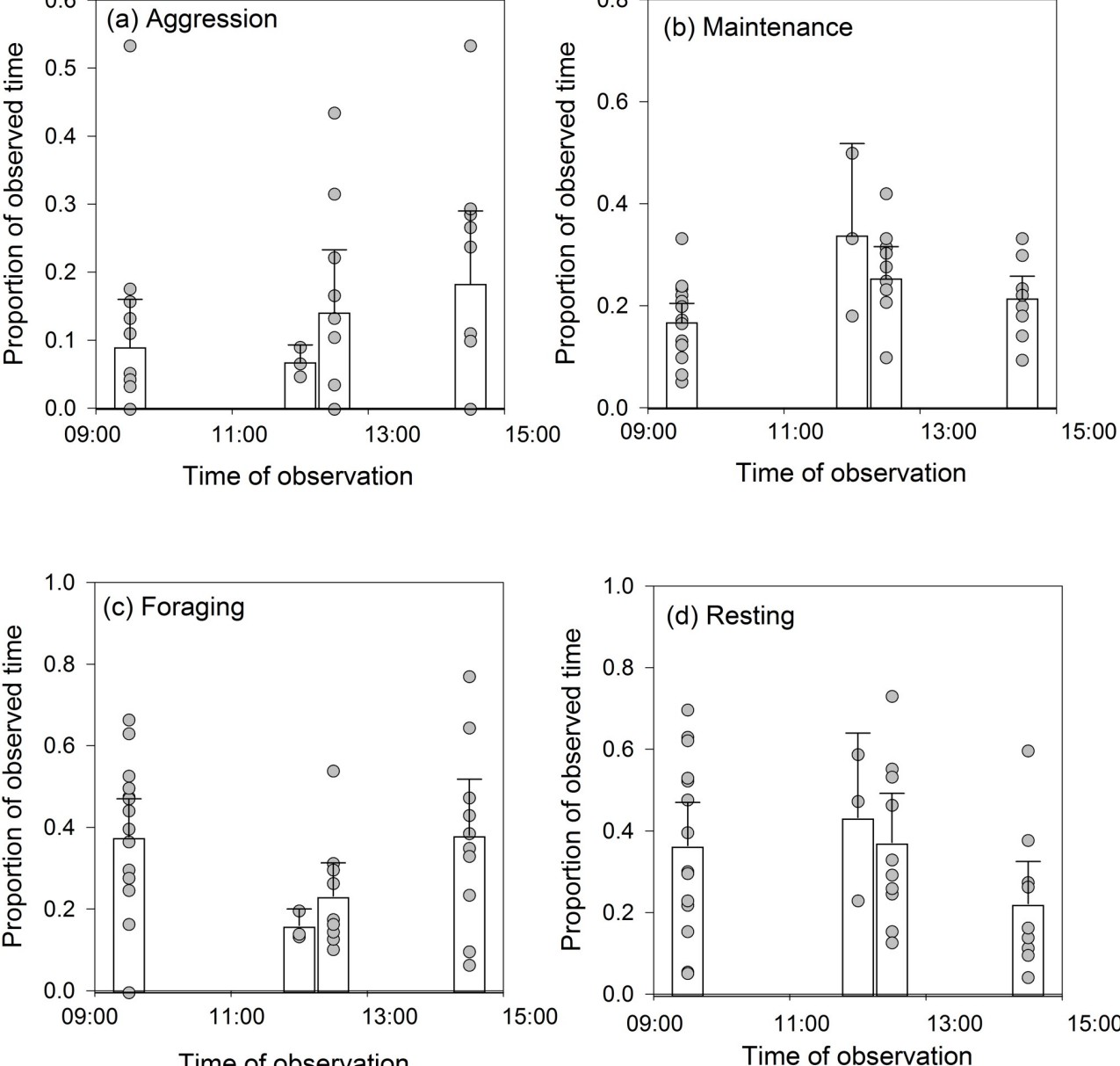

**Fig 3. Observed whooper swan behaviour at different times of day.** The bars represent the mean ± 95% CI time spent on each behaviour, whilst the filled circles represent the individual data points.

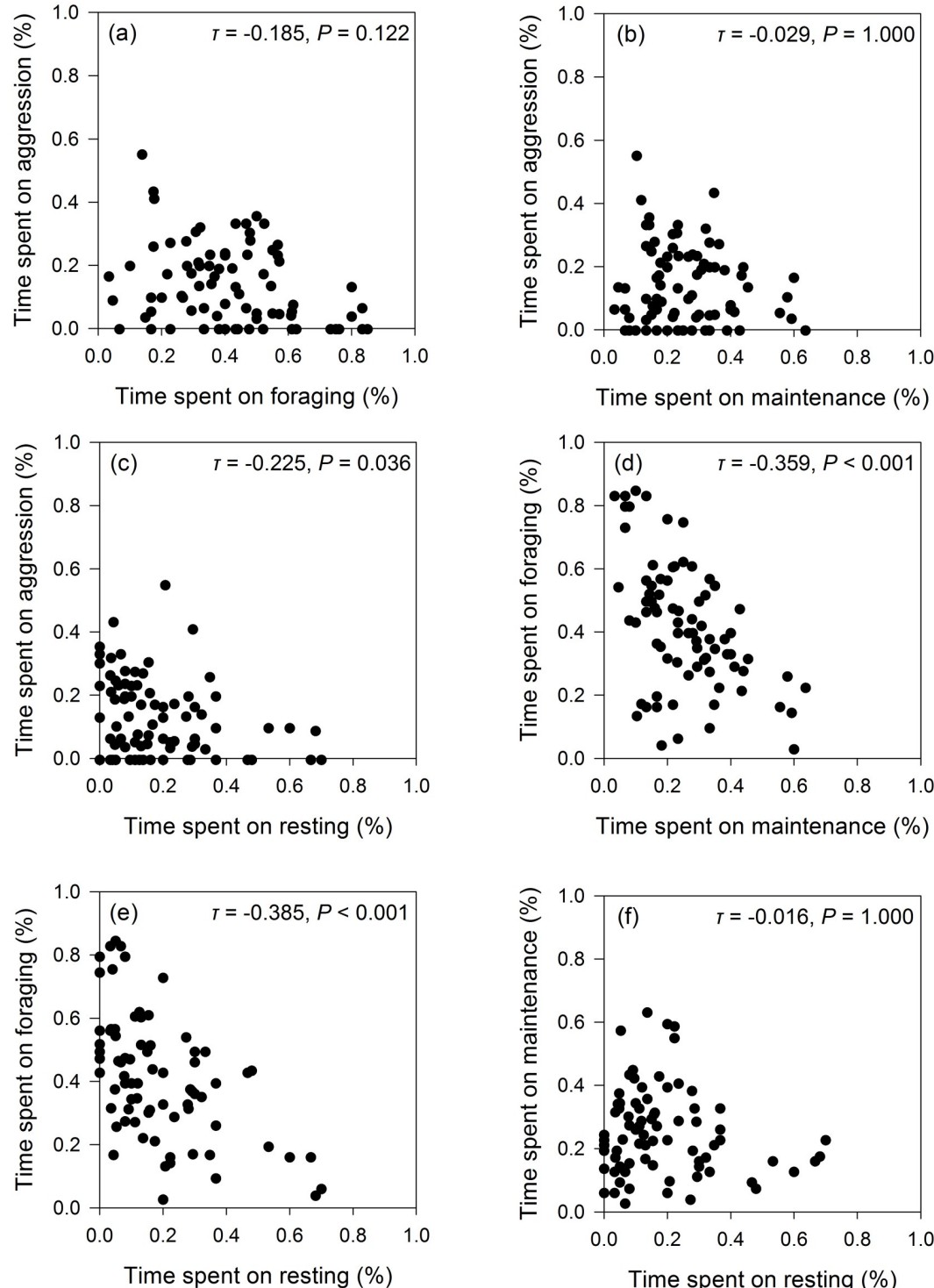

**Fig 4. Mute swan behavioural trade-offs.** Kendall's tau correlations between each of the recorded mute swan behaviours.

both maintenance ($\tau$ = -0.359, $P < 0.001$; Fig 4) and resting behaviours ($\tau$ = -0.385, $P < 0.001$; Fig 4).

No statistically significant correlations were detected, however, for mute swans between the proportion of time spent on aggression and foraging ($\tau$ = -0.185, $P = 0.122$; Fig 4), aggression and maintenance ($\tau$ = -0.029, $P = 1.000$; Fig 4), or maintenance and resting ($\tau$ = -0.016, $P = 1.000$; Fig 4).

For our observations of whooper swans, we found that the proportion of time spent on aggressive interactions was negatively correlated with the proportion of time spent resting ($\tau$ = -0.337, $P = 0.036$; Fig 5), whilst the proportion of time spent on foraging behaviour was negatively correlated with the proportion of time spent on resting behaviour ($\tau$ = -0.415, $P = 0.003$; Fig 5).

We detected no statistically significant correlations between the proportions of time that whooper swans spent engaged in aggression and foraging ($\tau$ = -0.149, $P = 0.820$; Fig 5), aggression and maintenance ($\tau$ = 0.078, $P = 1.000$; Fig 5), foraging and maintenance ($\tau$ = -0.149, $P = 0.685$; Fig 5), or between maintenance and resting ($\tau$ = -0.190, $P = 0.593$; Fig 5).

## Discussion

In this study, we used remotely collected behavioural observations made from a live-streaming webcam to investigate trade-offs in the behaviour of two bird species, the mute swan and whooper swan. These remotely-collected data allowed us to test four predictions regarding swan behaviour, as set out in our introduction: (i) no correlation between the time spent on aggressive and foraging behaviours; (ii) a correlation between aggression and resting; (iii) a correlation between aggression and maintenance behaviours; (iv) negative correlations between all non-aggressive behaviours. The findings that we report here illustrate how birds can trade-off their time investments in mutually exclusive behaviours within their time-activity budgets.

The time spent on foraging by mute swans at our study site (mean ± 95% CI = 42.1 ± 4.3%) was similar to the 43.3 ± 2.6% reported by another recent study of mute swan behaviour in eastern England [9]. Indeed, our observed value was also similar to other studies of foraging effort during winter, which reported 48% at a rural site in Poland [42], 41% in Denmark [21], and 36% in Ireland [33]. In contrast, an earlier study of mute swan behaviour at another wintering site in Poland found that foraging accounted for c.20% of time [43], whilst mute swans wintering in Scotland spent 58% of time foraging [44]. Our finding that our focal whooper swans spent 32.3 ± 6.0% of their time-activity budget on foraging was similar to the 35% reported from a study in China [45] and the c.30% reported by an earlier study conducted at our study site [16]. Research from Ireland and eastern England reported slightly higher foraging efforts of 40% [46] and 43% [9], respectively, whilst observations of whooper swans wintering in Turkey reported only 12% of time spent on foraging [22]. Some caution is needed in comparisons of foraging effort between different studies, as the time spent on foraging depends on multiple factors including daily energy requirement, food biomass, food energetic content, as well as the energetic cost of foraging behaviour [47–50]. However, because our observed time investments in foraging were well within the range reported previously for both species by studies conducted during winter, our data suggest that swans not only roosting on our study lake and feeding elsewhere, as is sometimes observed for overwintering swan populations [51]. Similarly, an earlier tracking study of GPS-tagged whooper swans at our study site indicated that individuals fed and roosted on the reserve, as many remained there for extended periods of time [27]. Although the study observed some localised (<10 km) movements from

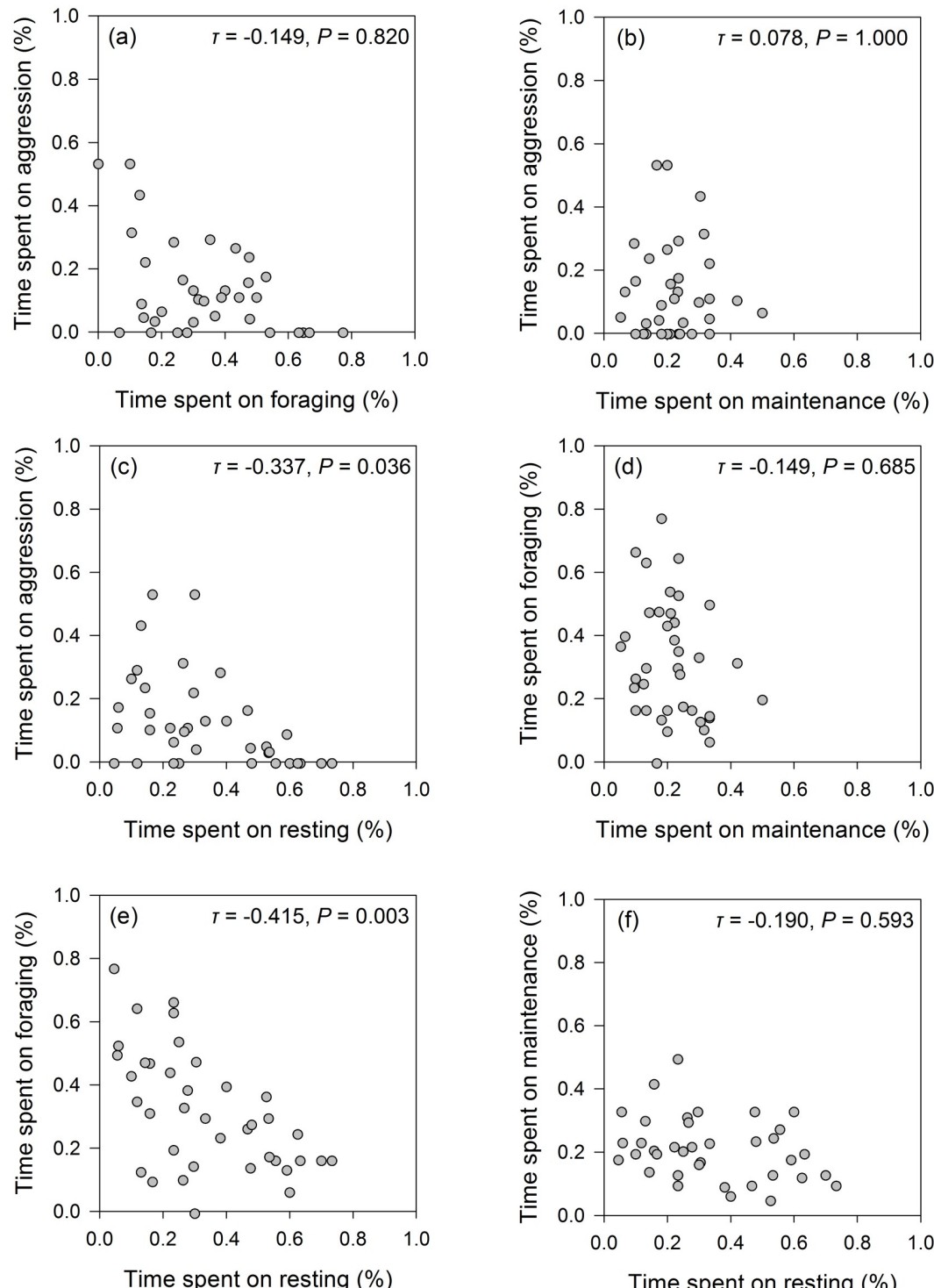

**Fig 5. Whooper swan behavioural trade-offs.** Kendall's tau correlations between each of the recorded whooper swan behaviours.

the reserve to adjacent areas, it did not find the systematic daily movements that would be expected if swans were only roosting on the reserve and were feeding elsewhere [27].

Maintenance behaviours such as preening were important activities for both swan species, accounting for 25% and 22% of the time activity budgets of our focal mute swans and whooper swans, respectively. These values are intermediate in comparison to previously published observations of wintering swans, which have ranged from 5% for mute swans wintering in Scotland [44] up to 44% for whooper swans wintering in Ireland [46]. Resting behaviour was also a major component of the time-activity budget of both focal species, accounting for 18% and 33% of time among mute swans and whooper swans, respectively. Again, these values appear intermediate when compared with previously published observations of wintering swans. As examples, whooper swans wintering in China spent 14% of time resting [45], whilst overwintering mute and whooper swans in eastern England spent 7% and 18% on resting, respectively [9]. In contrast, mute swans wintering in Denmark and Poland rested for 35% [21] and 46% [43] of their total time-activity budgets, respectively, whilst whooper swans wintering in Turkey spent 55% of their time resting [22].

Our observations that mute swans and whooper swans spent 14% and 13% of their time on aggressive encounters with other birds were higher than in many previous studies. As examples, whooper swans wintering in China spent 0.3% of time on aggression [45], mute swans in the USA spent 2% of time on aggression [20], whilst overwintering mute and whooper swans in eastern England spent 2% and 1%, respectively, of their time on aggression [9]. Our observed values were similar to the 16% recorded for mute swans during the breeding season in Poland [52]. However, studies of ecologically-similar large-bodied herbivorous avian species have recorded higher levels of aggression; for example, among barnacle geese (*Branta leucopsis*) up to 35% of time is spent engaged in aggressive interactions [53]. Indeed, a recent meta-analysis of aggression among waterbird species found that the time spent on aggression ranged from 0–35% [36], and so our observed values were within the range reported previously for waterbirds. Among swans, the time spent engaged in aggressive interactions has been shown to rise with increasing swan densities [9, 13]; therefore future work at our study site could assess whether the relatively high levels of aggression that were observed are due to higher densities of swans using the study lake.

Our observations of swan behaviour provided support for our first prediction, that there would be no trade-off between aggression and foraging. Aggressive behaviours represent a means of acquiring and maintaining access to food resources at the expense of competitors [24, 54]. Clustered food resources and the benefits of collective anti-predator behaviours mean that bird species often aggregate into flocks at feeding sites [55]. However, in such situations where multiple individuals seek to exploit shared food resources, those individuals that can dominate access to those resources will gain a competitive advantage. Previous research [56] has showed that as swan densities increased, competitively dominant individuals incurred smaller penalties to food intake rates compared with subdominant individuals. Aggressive behaviours are therefore frequently observed among birds such as swans at their feeding sites [36]. Indeed, among swans most aggression is directed towards conspecifics rather than heterospecifics, as the former represent the greatest competition for limiting resources such as food [9, 13, 20]. Individual swans which are actively foraging would therefore be expected to be the most likely to engage in aggression, whilst those not foraging would move away from feeding areas to minimise aggression [16].

Our findings also offered support for our second prediction, that there would be a trade-off between aggression and resting behaviours. In contrast, neither swan species showed any association between aggression and maintenance behaviours, thereby providing no support for our third prediction, that there would be a trade-off between these behaviours. In our two

focal species, increased aggression therefore appears to come at the expense of resting behaviour. Resting can be considered to represent a pool of time that can, at least to an extent, be redistributed to other activities as the individual requires. It is unclear how much time devoted to resting could be redistributed to activities such as aggression before there could be negative consequences on individuals; thus it is unclear what the ultimate limits of this behavioural trade-off might be. Aside from reducing the risk of predation [57, 58], periods of rest are known to be important in birds for health and cognition [59, 60]. To our knowledge there has been no research to date which has assessed the daily amount of rest that swans require to meet these needs. At our study site, mute and whooper swans spent on average 18.4% and 33.4% of their time engaged in resting behaviours. Given that previous studies have reported that swans can spend as little as 0.5% of their time resting [17], the redistribution of time from rest to other activities such as aggression is unlikely to be having marked impacts on our focal swans. Previous research has shown that male swans typically spend more time engaged in aggressive interactions compared with females [36]; however, sex cannot be readily determined visually for individual swans within a flock, and so the sex of our focal individuals could not be accounted for in our study. Previous research has found a balanced sex ratio among swans overwintering at our study site [16], and so any such sex effects were unlikely to lead to marked biases in our population-level data. Future research based on individuals of known sex could expand upon our findings by testing for trade-offs within each sex separately.

We found mixed support for our fourth prediction, that there would be a trade-off between all non-aggressive behaviours. As mutually exclusive activities, foraging and resting represent distinct modes of behaviour. Foraging is an activity that allows individuals to acquire the energy and nutrients needed to survive and reproduce, but exposes them to mortality risks such as predation, whereas resting allows individuals to minimise energy expenditure and predation risk [57]. Individual birds regulate their foraging effort to trade-off starvation and predation risk [58]. For animals such as swans that may fly between feeding sites, the mortality risk associated with foraging extends beyond predation to include flying accidents such as collisions with natural or man-made objects [61, 62]. Given the need for animals to achieve optimal trade-offs between foraging and resting, it is perhaps unsurprising that the strongest negative associations for both swan species were detected between foraging and resting.

Whilst we found no association between the durations of time spent on maintenance and resting, a trade-off between foraging and maintenance was detected only among mute swans. The biological significance of this disparity is unclear, especially as it is difficult to infer mechanistic explanations for behaviour from short observations. We quantified the behaviours of our focal swans over relatively short durations of 15 minutes, in common with most other studies of swan activities [9, 17, 20, 33, 34]. Whilst longer observation periods would have increased the information gained from each individual, they would also have increased the chance that the individual would have moved out of view during the observation. As a consequence, we were only able to assess for potential trade-offs over these short time periods. An unresolved question is therefore whether such trade-offs may persist over longer durations, or whether individuals alter their behaviour to undertake behaviour that was previously avoided. For example, where swans redistribute potential resting time to engage instead in aggressive interactions, do they later undertake additional resting behaviour to account for the earlier trade-off? Or is that potential resting time lost permanently? Such questions can likely only be addressed by quantifying swan behaviour over much longer time periods. The current widely used visual-based methods of quantifying swan time-activity budgets are likely to be unsuitable for longer term studies, as swans frequently move out-of-view. Alternative methods, such as the construction of time-activity budgets from data from individuals fitted with accelerometers

and other devices that allow behavioural activities to be inferred [50], may be needed to address these further questions.

Our work demonstrates how remotely-collected data can be used to investigate fundamental questions in behavioural research. Such remote data collection provides a number of advantages to behavioural scientists, including reduced impacts of disturbance on focal animals, reduced carbon footprint associated with repeated visits to observation sites, and greater accessibility for scientists who cannot physically travel to study sites [11–15]. Remote methods can also offer a means to collect data during the Covid-19 pandemic, which has curtailed the ability of researchers to visit field sites to undertake traditional methods of in-person data collection [63]. Given these advantages, we expect that remote methods of data collection will become an increasingly valued tool for behavioural research.

Observational studies have confirmed that animals are flexible in the amount of time that they spend on different behaviours, and as a consequence populations can exhibit high variation in the amount of time spent on different activities [64]. Such variation may reflect trade-offs in behavioural activities, whereby animals prioritise certain behaviours, depending on their current state and environmental conditions [10]. Our observations of two overwintering swan species are consistent with the idea that birds can trade-off their time investments in mutually exclusive behaviours within their time-activity budgets, at least over short time periods. For both of our study species, negative associations between foraging and resting, and between resting and aggression, suggest that swans can trade-off time investment in these behaviours. However, we also recognise that it is difficult to draw broader conclusions about the implications of such patterns of behaviour, including fitness impacts, from short-term observations conducted over periods of 15 minutes. Future research that undertook behavioural observations of known individuals over longer time periods, would further improve our understanding of the capacity of birds to trade-off behavioural activities within their time-activity budgets. The patterns of behaviour that we have documented here can inform the development of hypotheses regarding behavioural trade-offs in such studies.

## Acknowledgments

We are grateful to the staff at WWT Caerlaverock for the maintenance of the webcam used in this study. We thank Dr Claudia Mettke-Hofmann and an anonymous reviewer for their constructive comments on our manuscript.

## Author Contributions

**Conceptualization:** Kevin A. Wood, Paul E. Rose.

**Data curation:** Rebecca Lacey.

**Formal analysis:** Kevin A. Wood.

**Investigation:** Rebecca Lacey.

**Methodology:** Paul E. Rose.

**Project administration:** Paul E. Rose.

**Supervision:** Paul E. Rose.

**Visualization:** Kevin A. Wood.

**Writing – original draft:** Kevin A. Wood.

**Writing – review & editing:** Kevin A. Wood, Rebecca Lacey, Paul E. Rose.

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
