## [Decision Letter · Decision Letter 0]

30 May 2022

PONE-D-22-04625Assessing trade-offs in avian behaviour using remotely collected data from a webcamPLOS ONE

Dear Dr. Rose,

Thank you for submitting your manuscript to PLOS ONE. After careful consideration, we feel that it has merit but does not fully meet PLOS ONE’s publication criteria as it currently stands. Therefore, we invite you to submit a revised version of the manuscript that addresses the points raised during the review process.

The manuscript is well written and makes interesting use of webcam data. Both, the reviewer and I would like to see the following comments addressed before the manuscript can be considered for publication in PLOS ONE. A) Provide more background information about the species and similar studies that lead to the aims and predictions. Predictions may need to be reformulated. B) Consider including more data covering the entire day to control for any species-specific use of the reserve across the day. The way of data collection (webcam) would support this wider approach. C) Provide a more detailed discussion better fitting to the scope of the data.

All comments should be considered in the revision.

We look forward to receiving your revised manuscript.

Kind regards,

Claudia Mettke-Hofmann, PhD

Section Editor

PLOS ONE

**Journal requirements:**

2. In your Methods section, please provide additional information regarding the permits you obtained for the work. Please ensure you have included the full name of the authority that approved the field site access and, if no permits were required, a brief statement explaining why."

KLC (Straive) 15 Feb 22: Please provide additional details regarding participant consent. In the ethics statement in the Methods and online submission information, please ensure that you have specified what type you obtained (for instance, written or verbal, and if verbal, how it was documented and witnessed). If your study included minors, state whether you obtained consent from parents or guardians. If the need for consent was waived by the ethics committee, please include this information.

5. Please amend your list of authors on the manuscript to ensure that each author is linked to an affiliation. Authors’ affiliations should reflect the institution where the work was done (if authors moved subsequently, you can also list the new affiliation stating “current affiliation:….” as necessary).

**Additional Editor Comments:**

Comments by the Academic Editor:

Abstract:

Lines 32-34: Aggression and foraging are not negatively correlated. But are they positively correlated justifying the interpretation that aggression serves to secure access to food?

Lines 36-40: There is no trade-off between maintenance behaviour and resting for migratory whooper swan as it may be important to maintain good feather quality. Were there other trade-offs e.g. maintenance vs aggression? Is there more maintenance behaviour in whooper swans than mute swans justifying the interpretation?

Introduction:

Lines 54pp: This is interesting but deviates from your study as this links to individual differences, whereas your study is about differences between species. Please provide background information about differences in time budgets between species, ideally closely related species and also in relation to migration and residency to lead to your aim and selection of behaviours studied. The background should also include information about which behaviours are often traded-off against each other (e.g. vigilance-foraging, aggression-foraging, resting-foraging etc). Again, this then justifies the selection of behaviours and predictions for your study. You should also provide information about any known differences in the selected behaviours between the two focal species.

Methods:

Lines 128pp: Swans were randomly selected using a grid and random number generator. Were all grids selected at a comparable rate or was there e.g. a clumping in the middle? I ask as individuals may be quite stationary and any random clumping could lead to sampling a particular individual more often than others. Also, were only adult swans sampled or also juveniles?

Did you observe both swan species on the same day and time slot or how did you distribute observations?

Results:

Please provide all results even when they are not significant.

Discussion:

Lines 212pp: It would be good to repeat the predictions to remind the reader.

Lines 246pp: Potential sex effects. I am sure you know the sex ratio on the reserve for both species. I assume that usually entire families overwinter together, but it would be nice to have this mentioned somewhere.

Lines 278pp: You may also discuss that the birds may perform some particular behaviours on the lake you monitored (e.g. resting), whereas they may move somewhere else to perform other behaviours (e.g. foraging in the estuary). To which extend they move away for foraging may also differ between the two species. This should be discussed.

Lines 306pp: You mention individual differences here and that individuals trade-off behaviours. However, as far as I understand you were not able to identify individuals and your results reflect a population average rather than differences between individuals.

Fig. 1: Summing up the different behaviours seems to result in a value over 1 for both species. Wouldn’t you expect that the proportion of behaviours within each species adds up to 1 or below when you consider that there may be other behaviours and out-of-site? It definitely should not be above 1.

Reviewers' comments:

Reviewer's Responses to Questions

**Comments to the Author**

1. Is the manuscript technically sound, and do the data support the conclusions?

Reviewer #1: Partly

2. Has the statistical analysis been performed appropriately and rigorously? 

Reviewer #1: No

3. Have the authors made all data underlying the findings in their manuscript fully available?

Reviewer #1: Yes

4. Is the manuscript presented in an intelligible fashion and written in standard English?

Reviewer #1: Yes

5. Review Comments to the Author

Reviewer #1: The manuscript PONE-D-22-04625 is fairly well written but would be stronger if the authors (1) included more background information on the use of the reserve wetlands of the two focal species (2) conducted focal observations during a broader range across daylight hours of the day (3) kept conclusions regarding trade-offs in behaviour to the scope of the data as 15 minute intervals are ok to show that differences exist, but the implications of those differences likely have little biological relevance at that timescale for an individual. The authors did do a nice job of pointing out some of the benefits of the web cam approach for observations but missed an opportunity to take advantage of a constant ability to collect data across daily time periods via this type of data stream. A major revision to the paper as written to address these primary topics would be warranted prior to publication. Please see review file for specific notes on the manuscript.

6. PLOS authors have the option to publish the peer review history of their article (what does this mean?). If published, this will include your full peer review and any attached files.

Reviewer #1: No

---

## [Author Response · Author response to Decision Letter 0]

16 Jun 2022

Dear Professor Chenette,

 My co-authors and I have submitted a revision to our manuscript, “Assessing trade-offs in avian behaviour using remotely collected data from a webcam” (PONE-D-22-04625), as requested. We would like to thank the section editor Dr Claudia Mettke-Hofmann, as well as the anonymous reviewer, for their positive responses and constructive comments on the earlier version of our manuscript. We have revised our manuscript to address each of these comments and feel that our study has been greatly improved. In this letter of response, we present our detailed responses to each of the comments and discuss the changes we have made. Furthermore, we have now uploaded our data, as well as the code used in our analysis, to a figshare repository which is publically accessible. We would be grateful if the Data Availability statement could be updated to reflect this. Both the data files and the analytical code used in our study can be accessed via the following DOI: https://doi.org/10.6084/m9.figshare.20063225

Should you have any queries please do not hesitate to contact me.

 Yours faithfully,

Dr Paul Rose

 

For clarity, the comments from the reviewers are presented in bold and our responses are in italics. We have also indicated in our responses the line number(s) in the revised version of the manuscript where the relevant changes have been made.

Section Editor’s comments:

1. The manuscript is well written and makes interesting use of webcam data. Both, the reviewer and I would like to see the following comments addressed before the manuscript can be considered for publication in PLOS ONE. A) Provide more background information about the species and similar studies that lead to the aims and predictions. Predictions may need to be reformulated. 

– Our response: We are grateful for this positive and constructive feedback; revising our manuscript to address these comments has helped us to improve our work. In this letter we have responded to each of the comments in turn, detailing the changes that we have made. On the specific point raised in this comment, we have amended our introduction to provide greater background information on the behaviour of our two focal species (lines 77-92). This information is supported by citations of relevant previous studies of the behaviour of our two focal swan species (e.g. Holm 2002; Rees et al. 2005; Tatu et al. 2007; Włodarczyk 2017; Nergiz 2019; all of which are now cited in our revised manuscript). This new information has been integrated with our study predictions, so that the predictions presented in our revised manuscript are justified by the information that we provide (lines 77-106). We also more clearly draw the reader’s attention to key knowledge gaps that we intended our study to address (e.g. lines 90-92). Our revised manuscript therefore now features a much stronger link between our study predictions and the background information on the behaviour of both swan species.

2. B) Consider including more data covering the entire day to control for any species-specific use of the reserve across the day. The way of data collection (webcam) would support this wider approach.

– Our response: Please see our response to Reviewer 1 Comment 7. In our revised manuscript we now present new analyses of each behaviour across all of the different times of day at which observations were conducted. With these data and additional analyses, we now show that there are no temporal patterns in the behaviour of either swan species that could have confounded our original analysis (in particular, the pooling of data across time-periods).

3. C) Provide a more detailed discussion better fitting to the scope of the data.

– Our response: We have amended the discussion section in several ways to improve the text. New text has been added to discuss the behavioural findings for both species (lines 300-357). We have now included new text which provides better ecological context for our findings by comparing our findings with comparable previous studies of swan behaviour (lines 300-357). We now also provide the reader with information from previous studies on known factors which influence the amount of time spent on key behaviours, such as foraging (lines 312-316) and aggression (lines 353-357). Furthermore, we have added text to discuss evidence from tracking research that swans use the reserve for extended periods of time, and hence do not systematically move out of the reserve to feed (lines 320-325). Please also see our response to Reviewer 1 Comment 11 for details of additional revisions that we have made to our discussion.

4. Lines 32-34: Aggression and foraging are not negatively correlated. But are they positively correlated justifying the interpretation that aggression serves to secure access to food?

– Our response: On reflection we accept that the abstract focused too much on interpretations and not enough on reporting our findings. We have therefore revised the abstract to report all of the pairwise correlations between behaviours, even those that were non-significant (lines 32-39). Where correlations were detected, we have now stated whether these were positive or negative. We have also removed the interpretative text regarding access to food from the abstract, as we accept that these were not well-supported.

5. Lines 36-40: There is no trade-off between maintenance behaviour and resting for migratory whooper swan as it may be important to maintain good feather quality. Were there other trade-offs e.g. maintenance vs aggression? Is there more maintenance behaviour in whooper swans than mute swans justifying the interpretation?

– Our response: On reflection we accept that the abstract focused too much on interpretations and not enough on reporting our findings. We have therefore revised the abstract to report all of the pairwise correlations between behaviours, even those that were non-significant (lines 32-39). We have also removed the interpretative text from the abstract, as we accept that these were not well-supported.

6. Lines 54pp: This is interesting but deviates from your study as this links to individual differences, whereas your study is about differences between species. Please provide background information about differences in time budgets between species, ideally closely related species and also in relation to migration and residency to lead to your aim and selection of behaviours studied. The background should also include information about which behaviours are often traded-off against each other (e.g. vigilance-foraging, aggression-foraging, resting-foraging etc). Again, this then justifies the selection of behaviours and predictions for your study. You should also provide information about any known differences in the selected behaviours between the two focal species.

– Our response: As suggested, we have amended the text here to include mention of differences between species and between populations, rather than focusing on individuals. We now state that time-activity budgets can be a powerful tool for assessing differences in the behaviour of different animal species (lines 55-58). Our revised introduction includes much more background information on the types of behaviour shown by our two focal species (lines 77-92). We now mention that, where our two focal species co-exist, they show similar, but not identical, patterns of behaviour (lines 77-78); we provide additional ecological context for these two species (including their migratory/residency status) in the section on our study system (lines 114-122). We also present information on which behaviours may be traded off against others (lines 87-92), although we highlight that there has been little research on this topic, and so our study is a timely attempt to address this knowledge gap.

7. Lines 128pp: Swans were randomly selected using a grid and random number generator. Were all grids selected at a comparable rate or was there e.g. a clumping in the middle? I ask as individuals may be quite stationary and any random clumping could lead to sampling a particular individual more often than others. Also, were only adult swans sampled or also juveniles?

– Our response: We have added text to the methods section to confirm that, because swans were observed to use all grid cells (as well as all parts of the study lake), we see no reason why any individual grid cell would have been sampled more frequently than others (lines 163-166). Furthermore, we have now added text to explain that, because there were markedly fewer juveniles at the study sites, our observations were made of adult birds (lines 162-163). Our revised manuscript therefore now includes these important pieces of information to help the reader better understand our sampling approach.

8. Did you observe both swan species on the same day and time slot or how did you distribute observations?

– Our response: We have amended the text in our methods section to clarify that by randomly selecting the individual swan that was closest to the centre of randomly-generated grid co-ordinates, we effectively randomised the selection of swan species as well as the selection of an individual, because the individual that was closest to the centre of that grid could have been either a mute swan or a whooper swan (lines 161-162). This approach allowed us to gain behavioural data on both swan species without clumping of data on one species on certain days or time-periods.

9. Please provide all results even when they are not significant.

– Our response: As suggested, we have now included all behavioural results within our results section, including all non-significant findings, for both focal species (lines 268-271 and 281-285). Furthermore, we have also added text to report the non-significant correlations in the abstract (lines 32-39).

10. Lines 212pp: It would be good to repeat the predictions to remind the reader.

– Our response: We have followed the recommendation and have repeated the four predictions at the start of the introduction (lines 293-298). We also point the reader to the full rationale for these predictions, which can be found in the introduction.

11. Lines 246pp: Potential sex effects. I am sure you know the sex ratio on the reserve for both species. I assume that usually entire families overwinter together, but it would be nice to have this mentioned somewhere.

– Our response: We agree that it is useful to clarify these points. We have therefore amended our text to mention, with an appropriate supporting reference, that previous research has found a balanced sex ratio among swans overwintering at our study site (lines 126-127, 396-397). We also discuss how future research could investigate possible sex effects on behaviour, building upon our study (lines 399-400). Furthermore, we have also added text to confirm that whooper swan family groups typically remain together during winter, although this is somewhat less common among mute swans, for which family groups may break up sooner (lines 127-130). 

12. Lines 278pp: You may also discuss that the birds may perform some particular behaviours on the lake you monitored (e.g. resting), whereas they may move somewhere else to perform other behaviours (e.g. foraging in the estuary). To which extend they move away for foraging may also differ between the two species. This should be discussed.

– Our response: We agree that this is an important point that deserved greater attention in our manuscript. Therefore, we have added additional text to the discussion and methods to address this point. In our revised discussion we now compare the percentage of time spent on each behaviour with those reported in previous studies (lines 300-357). If our focal individuals had, for example, only been roosting on our study lake and had been undertaking daily foraging flights to feed elsewhere, then our data would show much higher resting and much lower foraging than in comparable studies. However, for all behaviours the time spent by our focal swans is well within the range reported in previous studies, and indicates that our focal swans were exhibiting their range of behaviours. For example, we now report that the 42% of time spent on foraging by mute swans at our study site was similar to the 43% reported for mute swans in eastern England [Wood et al. 2021], 48% at a rural site in Poland [Józkowicz & Gorska-Klek 1996], 41% in Denmark [Holm 2002], and 36% in Ireland [Keane & O'Halloran 1992]. In addition, it is also clear from previous research (Griffin et al. 2010; now cited in our study) that tracked individual swans at the study site that individuals fed and roosted on the reserve, as many remained there for extended periods of time. Although they observed some localised (<10 km) movements from the reserve to adjacent areas, they did not observe the systematic daily movements that would be expected if swans were only roosting on the reserve and were feeding elsewhere ; we now mention this study in our discussion (lines 320-325), as well as in our methods section in the text on the study system (lines 119-121). Overall, the evidence from our own work, and previous studies, suggests it is unlikely that the swans were performing only certain behaviours on our study lake and other behaviours unobserved away from our web cam; as described above, our revised manuscript now discusses this evidence.

13. Lines 306pp: You mention individual differences here and that individuals trade-off behaviours. However, as far as I understand you were not able to identify individuals and your results reflect a population average rather than differences between individuals.

– Our response: We accept the reviewer’s point and so we have removed the focus on individual animals from this paragraph, instead referring to populations (lines 446-463).

14. Fig. 1: Summing up the different behaviours seems to result in a value over 1 for both species. Wouldn’t you expect that the proportion of behaviours within each species adds up to 1 or below when you consider that there may be other behaviours and out-of-site? It definitely should not be above 1.

– Our response: We have checked our data thoroughly and we confirm that the time values displayed in Fig 1 for Mute Swans (aggression = 0.141, foraging = 0.421, maintenance = 0.254, resting = 0.184) and Whooper Swans (aggression = 0.126, foraging = 0.323, maintenance = 0.217, resting = 0.334) each sum to exactly 1.0, as would be expected. To ensure that readers can accurately evaluate the data presented in Fig 1, we have amended Fig 1 to increase the number of y-axis labels on the graph; the value of the y-axis is now indicated every 0.05% rather than every 0.10%. We have also rewritten our results section to report the precise values for the time-activity budgets of each species (lines 228-237). 

Reviewer 1’s comments:

1. The manuscript PONE-D-22-04625 is fairly well written but would be stronger if the authors (1) included more background information on the use of the reserve wetlands of the two focal species.

– Our response: We are grateful to the reviewer for their positive response and for the constructive comments, which have helped us to improve our manuscript. We have responded to each of the reviewer’s comments in turn, detailing the changes that we have made. On this particular point regarding our focal species, we have added additional information on the two swan species at our study site, including their use of the reserve wetland for all behaviours (lines 116-119), as well as information on their behaviours (lines 77-92). Please see also our response to Section Editor’s Comment #12, which made a similar point.

2. conducted focal observations during a broader range across daylight hours of the day

– Our response: Please see our response to Reviewer 1 Comment 7, where we present new analyses of each behaviour across all of the different times of day at which observations were conducted, which addresses the point raised here.

3. kept conclusions regarding trade-offs in behaviour to the scope of the data as 15 minute intervals are ok to show that differences exist, but the implications of those differences likely have little biological relevance at that timescale for an individual.

– Our response: On reflection we acknowledge the reviewer’s point and so we have revised our text in order to more firmly ground our discussion and conclusions in our data. We accept that the paragraph on the possible biological significance of the species disparity in whether maintenance and foraging showed a negative correlation, was rather speculative given our short observation durations. We have therefore removed this paragraph and instead we acknowledge that the biological significance of this disparity is unclear, and further acknowledge that it is difficult to infer mechanistic explanations for behaviour from short observations (lines 414-424). We have also softened the tone in the final paragraph to better reflect the tentative nature of conclusions drawn from short-term observations (lines 446-463). We have included within our discussion a paragraph which highlights the short duration of our observation periods, and proposes how future research using longer observations could shed further light on the questions that we address in our study (lines xx-xx); please see our response to Reviewer 1 Comment #10.

4. Line 75-83: Following your line of reasoning here that aggression is a means of maintaining access to food resources, I would have hypothesized that birds that were more aggressive had better access to high quality food resources and therefore would need to spend less time feeding as a result.

– Our response: We certainly see the reviewer’s point and agree that, over longer time periods, swans that maintain access to food resources through aggression might spend less time feeding, as they could gain their required energy more efficiently from a high quality food resource. However, over the short time durations (such as the 15 minute observation periods used in our study), any time that is spent on aggression activities is time that cannot be spent on foraging. This would offset any time gained from access to food resources. We have amended the text justifying our prediction to clarify this point about time scales (lines 95-97), so that our revised manuscript sets out the rationale for our predictions more clearly.

5. Line 85-88: These predictions contradict my expectations for why/how aggressive behavior would be rewarded. If a bird that spends a lot of time being aggressive to other birds is rewarded by the richest feeding areas and can quickly obtain the resources it needs then its time spent feeding is less and it receives other benefits like increased survival as feeding can be a risky behavior. Not sure that it is that critical a point as just testing these hypotheses is interesting but the assumption of no correlation between aggression and feeding time stated on Ln 80 is not intuitive.

– Our response: Please see our response to the previous comment (Reviewer 1 Comment #4), in which we explain how we have amended the text of our manuscript to improve the justification associated with our study prediction. Moreover, we also agree with the reviewer’s point that the direction of our prediction would not have affected the result found in our study, as our analyses tested for all types of association between the two behaviours (both positive and negative correlations, as well as no correlations). 

6. Line 96-98: Curious if the migratory whooper swans use the reserve for all of their feeding and roosting needs while in the area or if they predominantly feed or roost somewhere outside the reserve?

– Our response: Please see our response to Section Editor’s Comment #12, which made a similar point. In our revised discussion we now compare the percentage of time spent on each behaviour with those reported in previous studies (lines 300-357). If our focal individuals had, for example, only been roosting on our study lake and had been undertaking daily foraging flights to feed elsewhere, then our data would show much higher resting and much lower foraging than in comparable studies. However, for all behaviours the time spent by our focal swans is well within the range reported in previous studies, and indicates that our focal swans were exhibiting their range of behaviours. In addition, it is also clear from previous research (Griffin et al. 2010; now cited in our study) that tracked individual swans at the study site that individuals fed and roosted on the reserve, as many remained there for extended periods of time. Although they observed some localised (<10 km) movements from the reserve to adjacent areas, they did not observe the systematic daily movements that would be expected if swans were only roosting on the reserve and were feeding elsewhere ; we now mention this study in our discussion (lines 320-325), as well as in our methods section in the text on the study system (lines 191-121). Overall, the evidence from our own work, and previous studies, suggests it is unlikely that the swans were performing only certain behaviours on our study lake and other behaviours unobserved away from our web cam; as described above, our revised manuscript now presents and discusses this evidence.

7. Line 168-169: While behaviour may not have varied significantly, I would argue that there was a suggestion of differences between them during different times of day and month – see figure from (13). I would really like to see a breakdown of behaviour by hour from sunrise to sunset to test this.

– Our response: We understand the reviewer’s concerns, and agree that it is important to demonstrate that the time spent on each behaviour did not vary markedly across our observation periods, as such temporal variability within the diurnal period would have meant that it would be inappropriate to pool the observations from different times of day for our main analysis. We have therefore followed the reviewer’s suggestion of assessing whether the time spent on each behaviour by each of the two swan species did vary significantly between the different time periods for which we have data. We used Kruskal-Wallis tests to determine whether there were statistically significant differences in the time spent by each swan species on each behaviour (aggression, maintenance, foraging, resting) between each of the four time periods at which we made our behavioural observations. As we were interested in differences between any time period, we modelled the time periods as factors rather than as a continuous variable. Our new analysis found no statistically-significant differences in any behaviour between the four times-of-day at which sampling occurred, for either swan species. We have updated our manuscript to include this new analysis, in particular with new text in the methods (lines 193-205) and results (lines 243-247,253-257). We have also added two new figures to illustrate the results (new Fig 2 and Fig 3). Whilst we did not have data for every hour of the day, because day-length was short during our winter study period, our observations did span from early morning to dusk. Our earliest observations were from 09:30 and our latest observations were from 14:30; because day-length was short during our winter observations, it would not have been possible to collect data consistently at times that were outside of this sampling window. Crucially, we have now shown that we did not confound our analysis by pooling our data across all four times-of-day. The new analysis that we have added will reassure readers that our decision to pool data did not influence our results or conclusions.

8. Line 176-178: Was it assumed that both species of swans were present on the reserve for the entire day? If not, could the whooper swans be feeding offsite and roosting on the reserve?

– Our response: Please see our responses to Section Editor’s Comment #12 and Reviewer 1 Comment #6, which also address this question.

9. Lines 267-268: The disparity could also be due to differences in behavior by time of day between the species. If the more mobile whooper swans are flying out to other areas for various resources, the behaviour observed in the reserve may be biased. If there is movement data associated with the flocks or individuals it would be helpful to present for a clearer picture of their daily behaviour.

– Our response: The section referred to here by the reviewer has been deleted from the manuscript as part of our revisions in response to another comment (Reviewer 1 Comment #3). However, we agree that the point raised here regarding the potential movements of the birds, and how this could have impacted on our observational data, is an important one that we needed to address as part of our revisions. Therefore, we have added additional information from tracking studies and behavioural studies to provide the reader with information on this point. We now state that previous research (Griffin et al. 2010; now cited in our study) that tracked individual swans at the study site reported that individuals fed and roosted on the reserve, as many remained there for extended periods of time. Although they observed some localised (<10 km) movements from the reserve to adjacent areas, they did not observe the systematic daily movements that would be expected if swans were only roosting on the reserve and were feeding elsewhere ; we now mention this study in our discussion (lines 320-325), as well as in our methods section in the text on the study system (lines 119-121). In our revised discussion we now compare the percentage of time spent on each behaviour with those reported in previous studies (lines 300-357). If our focal individuals had, for example, only been roosting on our study lake and had been undertaking daily foraging flights to feed elsewhere, then our data would show much higher resting and much lower foraging than in comparable studies. However, for all behaviours the time spent by our focal swans is well within the range reported in previous studies, and indicates that our focal swans were exhibiting their range of behaviours. For example, we now report that the 42% of time spent on foraging by mute swans at our study site was similar to the 43% reported for mute swans in eastern England [Wood et al. 2021], 48% at a rural site in Poland [Józkowicz & Gorska-Klek 1996], 41% in Denmark [Holm 2002], and 36% in Ireland [Keane & O'Halloran 1992]. Overall, the evidence from our own work, and previous studies, suggests it is unlikely that the swans were performing only certain behaviours on our study lake and other behaviours unobserved away from our web cam; as described above, our revised manuscript now presents and discusses this evidence. We are grateful to the reviewer for prompting us to include this new information, which we believe has strengthened our manuscript.

10. Lines 278-295 This is a very nice summary of the limitations of the short duration observations and potential alternative approaches in the future.

– Our response: We thank the reviewer for their positive response to this section. We agree that it is important to discuss the limitations of the study, and so we have kept this section in our revised manuscript (lines 417-435).

11. Lines 313-317: Given data is collected in 15-minute blocks and the total budget for an individual for a given day would be a much better measure of trade-offs I would caution not to overstate the findings. Does a trade-off during a 15 minute bout have biological (impacts to fitness) for any individual or should those measurements be compared across a greater timescale before we draw conclusions regarding plasticity in behaviour.

– Our response: We agree that undertaking observations over longer time periods, such as determining time budgets for individuals for a given day, would provide more robust estimates of potential trade-offs between different behaviours; however, this was not an option for our study as we could not identify individuals from web cam footage and individuals typically do not remain visible ‘on screen’ for long enough to permit that approach. We have included text to explain these points (lines 419-423). We have also softened the tone in the final paragraph to better reflect the tentative nature of conclusions drawn from short-term observations (lines 446-463), and we have removed the mention of plasticity in behaviour. In particular, we have been careful not to attempt to draw any conclusions regarding fitness consequences. Instead, our revised conclusion acknowledges the interesting patterns that we have found were derived from short time periods of 15 minutes (lines 416-423, 456-458). We hope that the findings from our short-term study will inform future longer-tem studies, from which more robust conclusions regarding trade-offs can be drawn. Indeed, our study provides advice on how such studies could be carried out (lines 432-435, 458-461). We argue that the patterns of behaviour that we have documented in our study can inform the development of hypotheses regarding behavioural trade-offs in future longer-term studies.

– Our response: We have amended our manuscript to conform to PLOS ONE’s style requirements. On the title page, we have removed the postal code information from the affiliations listed for each author. The corresponding author’s initials are now given in parentheses after the email address. Within the manuscript, figures are now referred to using the abbreviation “Fig” and the corresponding number. Each figure now has a short title in bold before the main legend text. We have also renamed our files as recommended (for example, the file for figure 1 is now named “Fig1.tiff”.

2. In your Methods section, please provide additional information regarding the permits you obtained for the work. Please ensure you have included the full name of the authority that approved the field site access and, if no permits were required, a brief statement explaining why."

– Our response: We have amended the methods section to include a specific ethics subsection to provide this information (lines 183-190). This subsection states that our study was carried out with the prior approval of the ethics committee of the College of Life and Environmental Sciences of the University of Exeter (eCLESPsy002195). Moreover, we state that as our data collection was conducted virtually via a publically-accessible live-streaming webcam, no physical visits to the study site were undertaken, and hence no study site permits were required.

– Our response: We confirm that our study did not feature human participants and so participant consent was not required. Our manuscript reports an observational study of non-human animals. We have amended the manuscript to include an ethics section within the methods (lines 183-190), in order to explain this point.

– Our response: We have now uploaded our data, as well as the code used in our analysis, to a figshare repository, which is publically available. We would be grateful if the Data Availability statement associated with our manuscript could be updated to reflect that the data and code can be accessed via this DOI: https://doi.org/10.6084/m9.figshare.20063225

5. Please note that in order to use the direct billing option the corresponding author must be affiliated with the chosen institute. Please either amend your manuscript to change the affiliation or corresponding author, or email us at plosone@plos.org with a request to remove this option.

– Our response: The corresponding author, Dr Paul Rose, is affiliated with the University of Exeter, which is the institution selected for billing. We confirm that this affiliation is listed in our manuscript.

6. Please amend your list of authors on the manuscript to ensure that each author is linked to an affiliation. Authors’ affiliations should reflect the institution where the work was done (if authors moved subsequently, you can also list the new affiliation stating “current affiliation:….” as necessary).

– Our response: We confirm that the affiliations for all three authors are listed on the title page. For each author, the relevant affiliation is indicated with a superscript number, as per PLOS ONE’s guidelines.

---

## [Editor Report · Decision Letter 1]

28 Jun 2022

Assessing trade-offs in avian behaviour using remotely collected data from a webcam

PONE-D-22-04625R1

Dear Dr. Rose,

We’re pleased to inform you that your manuscript has been judged scientifically suitable for publication and will be formally accepted for publication once it meets all outstanding technical requirements.

Kind regards,

Claudia Mettke-Hofmann, PhD

Section Editor

PLOS ONE

Additional Editor Comments (optional):

All issues have been addressed.
---

## [Editor Report · Acceptance letter]

30 Jun 2022

PONE-D-22-04625R1 

Assessing trade-offs in avian behaviour using remotely collected data from a webcam 

Dear Dr. Rose:

I'm pleased to inform you that your manuscript has been deemed suitable for publication in PLOS ONE. Congratulations! Your manuscript is now with our production department. 

Kind regards, 

on behalf of

Dr. Claudia Mettke-Hofmann 

Section Editor

PLOS ONE